# Combined Transcriptome and Metabolome Analysis Reveals That Carbon Catabolite Repression Governs Growth and Pathogenicity in *Verticillium dahliae*

**DOI:** 10.3390/ijms252111575

**Published:** 2024-10-28

**Authors:** Yuan Wang, Di Xu, Boran Yu, Qinggui Lian, Jiafeng Huang

**Affiliations:** Key Laboratory of Oasis Agricultural Pest Management and Plant Protection Resources Utilization, College of Agriculture, Shihezi University, Shihezi 832000, China; wangyuan1@stu.shzu.edu.cn (Y.W.); xudi@stu.shzu.edu.cn (D.X.); yuboran@stu.shzu.edu.cn (B.Y.)

**Keywords:** carbon catabolite repression, *Verticillium dahliae*, metabolome, transcriptome

## Abstract

Carbon catabolite repression (CCR) is a common transcriptional regulatory mechanism that microorganisms use to efficiently utilize carbon nutrients, which is critical for the fitness of microorganisms and for pathogenic species to cause infection. Here, we characterized two CCR genes, *VdCreA* and *VdCreC*, in *Verticillium dahliae* that cause cotton Verticillium wilt disease. The *VdCreA* and *VdCreC* knockout mutants displayed slow growth with decreased conidiation and microsclerotium production and reduced virulence to cotton, suggesting that *VdCreA* and *VdCreC* are involved in growth and pathogenicity in *V. dahliae*. We further generated 36 highly reliable and stable *ΔVdCreA* and *ΔVdCreC* libraries to comprehensively explore the dynamic expression of genes and metabolites when grown under different carbon sources and CCR conditions. Based on the weighted gene co-expression network analysis (WGCNA) and correlation networks, *VdCreA* is co-expressed with a multitude of downregulated genes. These gene networks span multiple functional pathways, among which seven genes, including PYCR (pyrroline-5-carboxylate reductase), are potential target genes of *VdCreA*. Different carbon source conditions triggered entirely distinct gene regulatory networks, yet they exhibited similar changes in metabolic pathways. Six genes, including 6-phosphogluconolactonase and 2-ODGH (2-oxoglutarate dehydrogenase E1), may serve as hub genes in this process. Both *VdCreA* and *VdCreC* could comprehensively influence the expression of plant cell wall-degrading enzyme (PCWDE) genes, suggesting that they have a role in pathogenicity in *V. dahliae*. The integrated expression profiles of the genes and metabolites involved in the glycolysis/gluconeogenesis and pentose phosphate pathways showed that the two major sugar metabolism-related pathways were completely changed, and GADP (glyceraldehyde-3-phosphate) may be a pivotal factor for CCR under different carbon sources. All these results provide a more comprehensive perspective for further analyzing the role of Cre in CCR.

## 1. Introduction

Carbon catabolite repression (CCR) is a common mechanism used by microorganisms to coordinate the expression of the genes required for the preferential utilization of carbon sources [1]. The CCR mechanism allows microorganisms to accurately and efficiently utilize carbon nutrients under inconsistent nutritional conditions since it ensures the preferential utilization of easily metabolizable carbon sources such as D-glucose by repressing the expression of genes involved in the utilization of other alternative carbon sources [2]. In filamentous fungi, the expression of the genes encoding the polysaccharide-degrading enzymes α-amylase, cellulase, and hemicellulase, as well as other enzymes required for the utilization of carbon sources, are regulated by CCR, and as a result, the CCR system determines the hierarchy of carbon substrate utilization [2,3].

CCR has been extensively studied in the yeast *Saccharomyces cerevisiae* and the filamentous fungus *Aspergillus nidulans*. In *S. cerevisiae*, the DNA-binding repressor Mig1 is the main regulator of CCR and regulates the majority of the glucose-repressed genes [4]. In filamentous fungi, the CCR mechanism is regulated by a group of genes including *CreA*, *CreB*, *CreC*, and *CreD* [5]. The *CreA*-encoded protein contains two zinc fingers of the Cys2-His2 class, which are transcription factors that repress gene expression by binding directly to the 5′-SYGGRG-3′ motif in the promoters of the target genes [5,6]. For instance, the CreA of *A. nidulans* binds to the promoter region of xlnA and xlnD, which encode xylanase, and causes their direct repression; all the genes that are regulated by xlnR, which encodes the major inducer of xylanases, can be indirectly repressed by CreA [7]. In addition, CreA is a master regulator that governs diverse physiological processes including secondary metabolism, iron homeostasis, oxidative stress responses, development, N-glycan biosynthesis, the unfolded protein response, and nutrient and ion transport [8]. Ubiquitination ligases and deubiquitination enzymes interact with each other and control the transcription factors involved in the CCR mechanism. *CreB* encodes a deubiquitinase [9], and *CreC* encodes a WD40 protein, which together form a stable deubiquitination enzyme complex that eradicates ubiquitin moieties from CreA and other substrates, thereby stabilizing target proteins [10]. The *CreD*-encoded protein contains arrestin domains and PY motifs and has been reported to interact with the ubiquitin ligase HulA in *A. nidulans*; the CreD–HulA ubiquitination ligase complex helps in the ubiquitination of CreA, which opposes the CreB–CreC complex, ensuring the proper utilization of carbon sources for metabolism [11].

CCR has been widely reported to regulate fungal growth, secondary metabolism, and pathogenicity. The deletion of the *CreA* gene homolog *Cre1* in *Trichoderma reesei* resulted in an altered morphology, with smaller colonies having fewer aerial hyphae and spores compared with the parental strains, as well as reduced cellulase and hemicellulase production under repressing conditions [12]. MoCreA in *Magnaporthe oryzae* is required for vegetative growth, conidiation, appressorium formation, and pathogenicity [13]. The CreA of *A. fumigatus* is not required for pulmonary infection establishment, but it is essential for infection maintenance and disease progression; the loss of CCR inhibits fungal metabolic plasticity and the ability to thrive in dynamic infection microenvironments [14]. MoCreC in *M. oryzae* is involved in conidiation, growth, and pathogenicity; the deletion of *MoCreC* resulted in a reduced vegetative growth rate, lower conidiation production, impaired appressorium formation, and significantly decreased pathogenicity [15].

*Verticillium dahliae* is a soil-borne hemibiotroph phytopathogen fungus that causes vascular wilt in a wide variety of plants, including the economically important crops cotton, tomato, and sunflower [16]. The infection of roots by *V. dahliae* in soil leads to its colonization of vascular tissues in host plants, and it then produces microsclerotia during the progression of ripeness and senescence of the diseased plants. With the progressive decomposition of plant residue, the microsclerotia are released and can survive as resting structures in the soil for several years without a host plant, which makes *V. dahliae* difficult to control and eradicate [17].

Carbon is one of the most important nutrients for the growth and development of fungi. For most parts of its life in the host plant, *V. dahliae* is restricted to the xylem vessels of the vascular system; however, the xylem fluid provides an environment with limited carbon sources [18,19]. Studies have shown that *V. dahliae* encodes numerous carbohydrate-active enzymes including plant cell wall-degrading enzymes (PCWDEs), which degrade cell walls into metabolizable sugars and other mono- and oligomers for the fungus to utilize, allowing them to colonize plant xylem vessels [20]. In some fungi, the genes encoding these enzymes including cellulase, amylase, and xylanase are generally transcriptionally regulated by CreA-dependent CCR, but little is known about the functions of the genes involved in CCR in *V. dahliae*. Here, we show that VdCreA and VdCreC function as key components of carbon catabolite repression in *V. dahliae*, and disturbing CCR through knockout of *VdCreA* or *VdCreC* leads to significant defects in vegetative growth, conidiation, microsclerotium production, and pathogenicity. Further comparative transcriptomic and metabolomic analysis demonstrates that VdCreA and VdCreC are the regulators of CCR, and their disruptions and different carbon source conditions result in the extensive gene expression variations, including a large number of carbon metabolism enzymes, transcription factors, and genes encoding PCWDEs, yet they exhibit similar changes in metabolic pathways.

## 2. Results

### 2.1. Identification of VdCreA and VdCreC in V. dahliae

The two genes *VdCreA* and *VdCreC* were cloned and sequenced from the genomic DNA and cDNA of the V592 strain of *V. dahliae*, respectively. The *VdCreA* gene is composed of 1427 bp and contains an exon encoding 267 amino acids. The C-terminal part of the VdCreA protein has an acidic amino acid-rich region located at amino acids 110 to 116; and a highly conserved region at amino acids 117 to 161, which is identical between *A. nidulans*, *Fusarium oxysporum*, *Neurospora crassa*, and *Trichoderma reesei*; followed by a repressive region at amino acids 170 to 208 containing a nuclear export signal (NES), which has been shown to be essential for repression [21,22]. VdCreA lacks two N-terminal zinc finger domains, an alanine-rich region, and part of the nuclear location signal sequences compared to CreA proteins from other fungi (Figure 1A). The VdCreA protein shares the highest amino acid identity (99.63%) with that of *V. dahliae* VdLs.17, and it shares 66.33% and 51.67% amino acid identities with the CreA homologs in *V. longispore* and *Colletotrichum chlorophyti*, respectively.

*VdCreC* is composed of 2357 bp containing an open reading frame (ORF) that is interrupted by seven introns and encodes 611 amino acids. VdCreC protein contains a proline-rich (20/59 residues are proline) region near the N-terminus at amino acids 1 to 59. The C-terminal part of the protein contains a series of five WD40 (β-transducin-like) repeats at amino acids 260 to 300, 353 to 394, 395 to 436, 437 to 482, and 552 to 594 (Figure 1B). The VdCreC protein shares a 98.44% amino acid identity with that of *V. dahliae* VdLs.17, and it shares 80.41% and 72.68% amino acid identities with the CreC homolog in *V. longispore* and *Colletotrichum truncatum*, respectively.

Phylogenetic analysis based on amino acid sequences revealed that VdCreA and VdCreC were classified into two different clades, and they clustered with homologs of CreA and CreC from other fungi, respectively, which is consistent with their amino acid sequence similarity. This suggests that CreA and CreC in filamentous fungi were highly conserved during their evolution (Figure 1C).

### 2.2. VdCreA and VdCreC Knockout Leads to Serious Defects in Vegetative Growth and Conidiation

To confirm whether *VdCreA* and *VdCreC* participates in CCR and explore their biological role in the growth, development, and pathogenicity in *V. dahliae*, knockout mutants of *VdCreA* and *VdCreC* were generated via homologous recombination through which the targeted gene in the genome of the V592 strain was replaced with a hygromycin resistance cassette (hygromycin B phosphotransferase gene, *hph*), respectively (Appendix A). The two *VdCreA* knockout mutants (*ΔVdCreA-1* and *ΔVdCreA-2*) and the two *VdCreC* knockout mutants (*ΔVdCreC-1* and *ΔVdCreC-2*), the wild-type V592 strain, and the complementary strains or the overexpression strains were all cultured on PDA plates to compare their colony morphology and growth. *ΔVdCreA* displayed significantly more aerial hyphae, while *ΔVdCreC* exhibited decreased aerial hyphae (Figure 2A); the colony growth rate of both *ΔVdCreA* and *ΔVdCreC* were significantly lower than that of the V592 strain and their respective complementary or overexpression strains (Figure 2B,C). To further ascertain the functions of *VdCreA* and *VdCreC* in microsclerotial production, microsclerotia produced by each strain on cellophane overlaid on MM medium were weighed. The results showed that the knockout of *VdCreA* and *VdCreC* led to reduced microsclerotia in *V. dahliae*. Compared to the V592 strain, the microsclerotia were decreased by 1.7 times and 2.0 times in *ΔVdCreA* and *ΔVdCreC*, respectively; whereas the overexpression of *VdCreA* and *VdCreC* resulted in increased microsclerotia production by 1.7 times and 1.8 times, respectively (Figure 2D,E). The results suggested that *VdCreA* and *VdCreC* are involved in radial growth and microsclerotium production in *V. dahliae*.

The effect of the two genes’ knockout on conidium production was also investigated. Both *ΔVdCreA* and *ΔVdCreC* showed significant reduced conidiation. At 7 days post-inoculation (dpi), *ΔVdCreA* and *ΔVdCreC* produced about 2.0 times and 2.5 times less conidia than the V592 strain, respectively; the overexpression strains *OEVdCreA* and *OEVdCreC* produced about 1.5 times and 1.1 times more conidia than the V592 strain (Figure 3A,B). Consistent with these results, under the optical microscope, *ΔVdCreA* and *ΔVdCreC* hardly differentiated into verticillate conidiophore stalks, whereas the V592 strain, the complementary strain *ECVdCreA*, and the overexpression strains (*OEVdCreA* and *OEVdCreC*) produced many verticillate conidiophore stalks, on which conidia were gathered (Figure 3C,D); these results indicated that *VdCreA* and *VdCreC* are critical for the conidiogenesis of *V. dahliae*.

### 2.3. VdCreA and VdCreC Are Involved in Pathogenicity

To assess the roles of *VdCreA* and *VdCreC* in virulence, the susceptible cotton cultivar Junmian No. 1 was inoculated with conidial suspensions of the wild-type strain V592, *ΔVdCreA*, *ΔVdCreC*, and their complementary strains. *ΔVdCreA* and *ΔVdCreC* exhibited markedly reduced virulence in cotton plants compared with the V592 strain, which induced severe tissue chlorosis and wilt symptoms (Figure 4A). At 25 dpi, the disease index of plants infected by *ΔVdCreA* and *ΔVdCreC* was 41.7 and 63.9, respectively; whereas the disease index of plants infected by the V592 strain reached 91.0, and the disease index caused by the complementary strains of *ΔVdCreA* and *ΔVdCreC* was restored to 83.3 and 84.0, respectively (Figure 4B). Therefore, we concluded that the *VdCreA* and *VdCreC* genes are involved in fungal virulence.

### 2.4. VdCreA and VdCreC Are Essential for Carbon Catabolite Repression

To obtain evidence of *VdCreA* and *VdCreC* being involved in CCR in *V. dahliae*, all strains were inoculated on Czapek–Dox agar plates containing different sole carbon sources with or without glucose. Although in most cases the growth rates of *ΔVdCreA* and *ΔVdCreC* grown on these carbon sources were slower than that of the V592 strain, the relative growth rate of each mutant was significantly higher than that of the V592 strain, except for those grown on CMC and on CMC plus glucose (Figure 5A–C). Despite each strain displaying a growth defect on CMC plus glucose compared to the growth on CMC, *ΔVdCreA* was found to have higher cellulase activity based on the sizes of clear zones (halos) around their colonies (Figure 5D), where the de-repression rate of cellulase activity in *ΔVdCreA* grown on CMC and on CMC plus glucose was 55% and 60%, respectively; similar results were also obtained in *ΔVdCreC*, where the de-repression rate of cellulase activity grown on CMC and CMC plus glucose was 10% and 18%, respectively (Figure 5E). The amylase activity was also measured, in which the de-repression rate of amylase activity in *ΔVdCreA* grown on starch and starch plus glucose was 40% and 72%, respectively; the de-repression rate of amylase activity in *ΔVdCreC* grown on starch and starch plus glucose was 20% and 65%, respectively (Figure 5F). Furthermore, the de-repression rates of cellulase and amylase activity were significantly increased under glucose-added conditions, especially for amylase activity. The addition of glucose caused a remarkable increase in the de-repression rate; the fold increase was 1.8 for *ΔVdCreA* and 3.3 for *ΔVdCreC* compared to no glucose addition (Figure 5F). The results suggested that *VdCreA* and *VdCreC* participated in the CCR associated with cellulase and amylase in *V. dahliae*.

To further investigate the role of *VdCreA* and *VdCreC* in CCR, 2-deoxy-d-glucose (2-DOG) was used as a repressing carbon substrate to assess the glucose-induced repression through measuring the colony growth of the V592 strain and the two mutants grown on Czapek–Dox medium containing CMC plus 2-DOG, and starch plus 2-DOG. 2-DOG is a non-metabolizable D-glucose analog and can inhibit glucose metabolism in the cells [23]. The results showed that the growth repression rate was higher in the wild-type strain V592 than that in the *ΔVdCreA* and *ΔVdCreC* (Figure 5G,H). Thus, similar to many other filamentous fungi, *VdCreA* and *VdCreC* function as key regulators of CCR in *V. dahliae*.

### 2.5. Global Perspective of Gene and Metabolic Expression Profiles

To further gain insight into the VdCreA- and VdCreC-mediated regulation mechanisms, changes in the transcriptomics and metabolomics were assessed to compare between *ΔVdCreA*, *ΔVdCreC*, and the wild-type V592 strain under inducing conditions (starch and CMC) and repressing conditions (starch plus 2% glucose, and CMC plus 2% glucose). We constructed 36 transcriptome libraries for the V592, *ΔVdCreA*, and *ΔVdCreC* strains under different carbon sources, collecting an average of 42.17 million raw reads from each library (Appendix A). The reliability of the RNA-seq data was confirmed by qRT-PCR analysis of nine randomly selected genes (Appendix A).

A total of 12,834 unigenes were identified, and the expression level of each gene was calculated using FPKM (fragments per kilobase of exon model per million mapped fragments) and subjected to principal component analysis (PCA) (Figure 6A). PC1 and PC2 together explained 64.52% of the variance between expression profiles. Replicates of the same treatment showed high similarity and consistency, always clustering together with no obvious outliers. Starch or cellulose conditions seemed to be the main factors affecting gene expression changes in the strains, with samples from different carbon sources consistently distributed on both sides of PC1. *VdCreA* gene knockout also led to gene expression changes, exhibiting an obvious separation on both sides of PC2 between *VdCreA* knockout mutants and wild-type strains under repression conditions. Different treatments were grouped and compared pairwise to screen for differentially expressed genes (DEGs). We used an up-set plot to display the statistical results (Figure 6B). The VXD and AXD groups had the most DEGs, with 563 genes differentially expressed only in this group. The focus was on those DEGs that were always present in each group. Under repression conditions, compared with the wild-type strain, there were the same 271 DEGs in *ΔVdCreA* and *ΔVdCreC* (marked in red), of which 126 DEGs were present in all groups. These 271 DEGs may be the main genes involved in CCR. Transcription factors are usually the hubs of gene expression regulation. We identified all transcription factors (TFs) in the 271 DEGs and grouped them (Figure 6C). In comparison, there were more TFs under repressing conditions, mainly four *Zn-clus* (Zn2Cys6 TF), three *GNAT* (Gcn5-related N-acetyltransferases) TFs, and one *SNF2* TF shared by two groups.

The same batch of samples also underwent metabolomics analysis, detecting a total of 17,723 peaks and identifying 2986 metabolites. The TIC (total ion current) of the mixed sample was used to demonstrate the characteristics of the mass spectral peaks (Figure 6D,E). The main products in the positive ion mode were agroclavine, while in the negative ion mode, the main products were 5-δ-hydroxybutyl hydantoin. A total of 1672 (55.94%) metabolites were successfully annotated based on the public database, which can be classified into 122 classes and 214 subclasses. The classes with the highest number of identified metabolites are carboxylic acids and derivatives (254), fatty acyls (208), and organooxygen compounds (190). The subclasses with the most identified metabolites are amino acids, peptides, and analogs (231), carbohydrates and carbohydrate conjugates (119), and fatty acids and conjugates (99). We have uploaded the mass spectral characteristics, mass m/z values, and annotation results of all metabolites to facilitate the reuse of scientific data (Appendix A). Based on the expression levels of metabolites, we calculated the correlation between all samples (Figure 6F). Almost all biological replicates always cluster together in the heatmap clustering results, indicating that the detection of metabolism has good repeatability. However, VD2 is an exception, it is separated from the other two replicates, and in subsequent analyses, this sample was discarded. In addition, the correlation of metabolite spectra of standard products used for quality control also proved the stability of the detection results (Appendix A). After removing the metabolites with extreme expression in each group (the top 1% and bottom 1% of expression), we counted the top 50% of metabolites in terms of expression. Overall, the range of metabolite expression changes in all groups is very large. After repression, regardless of the carbon source, the expression levels of the main metabolites in the wild-type, *ΔVdCreA*, and *ΔVdCreC* strains all increased (Figure 6G,H). We used OPLS-DA (orthogonal partial least squares discriminant analysis) to perform pairwise comparisons of groups for supervised multivariate analyses to screen for differentially expressed metabolites (DEMs). We displayed the model evaluation results between the four groups of samples after repression (Figure 6I). The predictive principal component in all four groups explained more than 50% of the variance, and samples from different groups showed clear separation in the predictive principal component. The Q2Y value of the model exceeded 89%, and the R2Y value exceeded 99%, signifying substantial shifts in metabolic dynamics within mutants, and the model could stably and reliably explain the differences in metabolites between samples.

### 2.6. Weighted Gene Co-Expression Network Analysis (WGCNA)

The results of gene expression profiling and metabolite correlation heatmaps both indicated that different carbon source dependencies had a significant impact on the expression of genes and metabolites in the samples. To focus on *VdCreA* and *VdCreC*, we divided the gene expression profiling results under two carbon source conditions and performed WGCNA on all DEGs separately. Specifically, DEGs with an average expression level between samples less than 1 and a missing degree greater than 50% were first filtered out. Subsequently, a sample phylogenetic tree was constructed based on expression levels (Appendix A). Under starch conditions, one sample each from the VD and VDD groups was an outlier. Under cellulose conditions, one sample each from the VX and CX groups was an outlier. These samples were removed in subsequent analyses. Then, the most suitable soft threshold was selected based on a correlation exceeding 75% (Figure 7A,B), with 12 and 15 serving as the soft thresholds under starch and cellulose conditions, respectively, to construct the adjacency matrix. The minimum module gene number was uniformly set to 30. Ultimately, under starch conditions, 5558 DEGs were divided into 18 modules, with an average of 308 genes per module. The modules containing *VdCreA* and *VdCreC* had 1478 and 1597 genes, respectively (Figure 7C). Under cellulose conditions, 5278 DEGs were divided into 21 modules, with an average of 263 genes per module. The modules containing *VdCreA* and *VdCreC* contained 679 and 211 genes, respectively (Figure 7D). The connectivity of eigengenes in different modules and the degree of association of module genes with repressing conditions were calculated separately. Under starch conditions, a module of 37 genes had a 78% correlation with repressing treatment (*p* = 1.3 × 10^−8^) (Figure 7E). Under cellulose conditions, a module of 94 genes had an 83% correlation with repressing treatment (*p* = 4.7 × 10^−25^) (Figure 7F).

### 2.7. VdCreA Primarily Co-Expresses with Downregulated Genes

We further calculated the Spearman correlation coefficient between genes within each module and metabolites. Genes with a correlation greater than 90% that passed the significance test were considered co-expressed genes or metabolites. *VdCreA* co-expressed with 23 DEGs (Figure 8A), regardless of whether starch or cellulose was the carbon source. Among them, 22 were monotonically downregulated and 1 *TYW* (tRNA wybutosine-synthesizing protein) was monotonically upregulated. Among the monotonically downregulated genes, besides two involved in amino acid biosynthesis (*PYCR*, pyrroline-5-carboxylate reductase; *ALS*, acetolactate synthase), the functions of the remaining genes did not show obvious enrichment. Seven of these genes were hypothetical proteins, and there were also two zinc finger TFs (*ZF*, *C2H2*). At the same time, 37 DEMs were strongly correlated with *VdCreA* knockout. These DEMs were widely involved in various metabolic processes, mainly enriched in arachidonic acid metabolism (six DEMs), linoleic acid metabolism (five DEMs), and alpha-linolenic acid metabolism (three DEMs) pathways. The expression profiling results showed that these genes were mainly lowly expressed under repressing conditions, especially when starch was the carbon source (Figure 8B).

Further, we identified the conserved motifs in the promoters of 23 genes co-expressed with *VdCreA*, and a total of 10 motifs were identified across all genes (Appendix A). Two of these motifs (Motif 2 and Motif 5) were identified as SYGGRG-like motifs (Figure 8D). SYGGRG is considered the consensus sequence for *CreA* binding target genes [5], these two motifs are distributed in the proximal promoters of seven co-expressed genes (Figure 8C), including genes encoding PYCR, PSP1 (PSP1 domain-containing protein), nsLTPs (non-specific lipid-transfer protein), two zinc fingers (RING-3 and zinc finger), and two hypothetical proteins (VDAG_05397 and VDAG_07703). These genes may be regulated by *VdCreA*, functioning downstream. We focused on the changes in metabolites in each gene’s respective pathway (Figure 8D). PYCR, which participates in the arginine and proline metabolism pathway and catalyzes the interconversion between proline and 1-Pynoline-5-carboxylate, is highly expressed in *ΔVdCreA*, while the proline content in *ΔVdCreA* is extremely low. This suggests that VdCreA may target the regulation of PYCR expression, thereby mediating the conversion between proline and arginine and affecting the normal growth of the strain.

### 2.8. VdCreC Indirectly Engages in Multiple Metabolic Dynamics

In contrast, *VdCreC* was only strongly directly correlated with one hypothetical protein (*VDAG_09398*). We further screened 28 secondary network genes co-expressed with this gene (Figure 9A). After *VdCreC* knockout, under different carbon source conditions, 7 genes were monotonically downregulated, 1 gene was monotonically upregulated, and 22 genes exhibited no differential expression. These genes include *Exg* (exoglucanase) involved in cellulose hydrolysis, *CHD* (choline dehydrogenase) involved in choline metabolism, *PLB* (pectate lyase B) involved in pectin degradation, etc. In addition, 13 genes were annotated as hypothetical proteins, and their functions are largely unknown. In total, 84 metabolites were strongly correlated with *VdCreC* expression. Most of these metabolites were involved in the synthesis of various amino acids such as arginine, proline, and phenylalanine. The rest were enriched in secondary metabolite synthesis, glycolysis/gluconeogenesis, folate biosynthesis, and other pathways. The expression profiling showed that although the expression trends of these genes were not completely consistent, compared with the wild type, the overall expression level of these genes decreased after *VdCreC* knockout, regardless of the carbon source; when cellulose was the carbon source, the expression level of these genes was higher (Figure 9B). The findings imply that, despite the absence of co-expression between *VdCreC* and a multitude of functional genes at the gene expression level, and the lack of significant differential expression in a large number of genes within the co-expression network, it is nevertheless indirectly implicated in alterations in a multitude of pathways related to carbon metabolism.

### 2.9. Distinct Carbon Sources Induce Entirely Different Expression Networks

We extracted two modules related to repressing from the WGCNA results and performed KEGG enrichment on the modules’ genes (Appendix A). Among the top 20 enriched pathways, 4 were consistent: carbon metabolism, pentose phosphate pathway, arachidonic acid metabolism, and glutathione metabolism. Further, a similar process was used to construct the co-expression network of genes and metabolites within the module, with genes involved in carbon metabolism serving as hub genes (Figure 10A). Under starch conditions, the hub gene encoded a 6-phosphogluconolactonase, which is one of the core roles in the pentose phosphate pathway, and it was always upregulated after repression in all samples. There were 22 co-expressed genes with the hub gene, of which 21 were only upregulated, and 1 *PG* (polygalacturonase)-encoding gene was only downregulated. Under cellulose conditions, among the five hub genes, the genes encoding *G6PD* (glucose-6-phosphate 1-dehydrogenase) and glyoxylate reductase were always upregulated. *G6PD* is the rate-limiting enzyme of the pentose phosphate pathway, and glyoxylate reductase is involved in the synthesis of succinate in the glyoxylate cycle. One *PrpC* (2-methylcitrate synthase), one *MCL* (mitochondrial 2-methylisocitrate lyase), and one *2-OGDH* (2-oxoglutarate dehydrogenase E1) were always downregulated under repressing conditions. These three genes encode key enzymes in the citric acid cycle. There were 47 genes co-expressed with the hub genes, of which 25 genes were always upregulated, 21 genes were always downregulated, and 1 *DAAO* (D-amino acid oxidase) gene had opposite expression trends in different groups. Surprisingly, although the two gene expression networks were markedly distinct, the gene functions in the two modules overlapped and their strongly correlated metabolites were basically the same, suggesting that the strain may complete CCR through similar pathways. The genes in the two co-expression networks were strongly correlated with as many as 685 metabolites. Enrichment analysis showed that these metabolites were widely involved in lipid metabolism, nucleotide metabolism, amino acid metabolism, carbohydrate metabolism, etc., indicating that CCR could systematically affect the activity and efficiency of various biochemical processes. These metabolites were mainly enriched in carboxylic acids and derivatives (14 metabolites), organooxygen compounds (12 metabolites), prenol lipids (10 metabolites), and benzene and substituted derivatives (10 metabolites). Regardless of which module’s co-expressed genes had different preferences for carbon sources, according to the expression profile, they can generally be divided into two categories (Figure 10B). One category contains most genes, such as 6-phosphogluconolactonase, *PrpC*, *G6PD*, *2-OGDH*, and glyoxylate reductase, etc. Once the strain was in repressing conditions, regardless of the carbon source or strain, its expression level would stably increase or decrease. The difference was that some of them had higher expression in starch, and others had higher expression in cellulose. These genes may participate in the CCR process regardless of the carbon source. However, some other genes were only induced in one carbon source, such as *MCL*, which had almost no expression under starch conditions and was downregulated by repressing induction under cellulose conditions. *PG* was only downregulated under starch conditions. Whether they participated in CCR was related to the carbon source faced by the strain. The subtle differences in expression patterns between these co-expressed genes may be the handle for the strain to fine-tune CCR to efficiently utilize carbon sources. The core genes functioned as hub nodes within the expression network, offering a leverage point for further analysis of CCR.

### 2.10. The Glycolysis/Gluconeogenesis Pathway Is Entirely Altered

We integrated the expression profiles of genes and metabolites, focusing on the glycolysis/gluconeogenesis and the pentose phosphate pathway, which are considered to be the main pathways in the CCR process. The changes in the glycolysis/gluconeogenesis pathway were thorough (Figure 11), with 91 genes and 7 metabolites in this pathway detected to have differential changes. The core change was GADP (glyceraldehyde-3P), an intermediate product of glycolysis and one of the sugar products of the Calvin cycle. When starch was the carbon source and was subject to glucose repression, the content of GADP in the wild-type strain decreased, while the content of this substance greatly increased in *ΔVdCreA* and *ΔVdCreC*. In contrast, there was no obvious change in this substance when cellulose was the carbon source. One reason may be the upregulation of *FBA* (fructose-1,6-diphosphate aldolase) after *VdCreA* or *VdCreC* knockout. This enzyme catalyzes the cleavage of β-D-Fructose-1,6P_2_ into DHAP (Glycerone-P) and GADP, and was also only induced under starch conditions. The content of DHAP was also significantly different under different carbon sources. The difference was that the content of this substance in *ΔVdCreC* was higher. Two *TPI* (Triose phosphate isomerase) genes, which encode enzymes that catalyze the interconversion of DHAP and GADP, had significant differential expression in different groups and inconsistent expression trends. Another important metabolite that changed was oxaloacetate, a key intermediate in glycolysis and gluconeogenesis. In the samples with *VdCreA* or *VdCreC* knockout, under starch conditions, the content of oxaloacetate significantly decreased under CCR status, while its content was higher throughout when cellulose was the carbon source. The differential effects of the two carbon sources on pathway-related genes were common. Some *G6PI* (glucose-6-phosphate isomerase), *GPI* (glucose phosphate isomerase), *PK* (pyruvate kinase), and *acs* (acetyl-coenzyme A synthetase) genes only responded to cellulose carbon sources, while some *FBA*, *gpmB* (probable phosphoglycerate mutase GpmB), and *DLAT* (dihydrolipoamide S-acetyltransferase) genes only responded to starch carbon sources. In addition, a large number of genes encoding the *ALDH* (aldehyde dehydrogenase), *adhP* (10-acetyl-3,7-dihydroxyphenoxazine), and *adh* (alcohol dehydrogenase) enzymes had differential expression and large differences in expression patterns. These enzymes catalyze the conversion between acetate, acetaldehyde, and ethanol, although their contents had not been detected to have differential changes. As anticipated, the impact of CCR on glycolysis/gluconeogenesis was significant, exhibiting variances between the two carbon sources. Irrespective of the carbon source, GADP and oxaloacetate seemed to be of particular importance, potentially serving as pivotal elements of CCR, attributable to coordinated alterations in their metabolic and genetic strata.

### 2.11. The Pentose Phosphate Pathway Experiences Significant Impact

Differential expression was detected in 37 genes and 6 metabolites in the pentose phosphate pathway (Figure 12). Under starch conditions, the content of 2-Deoxy-D-ribose was lower, and the knockout of *VdCreA or VdCreC* and repression conditions further caused its content to decrease. However, under cellulose conditions, this substance had a higher content in all types of strains and was hardly affected by repression. Ribokinase (rbsK) catalyzes the phosphorylation of 2-Deoxy-D-ribose to form DR5P (D-Ribose-5P), and its expression only changed under cellulose conditions. Although DR5P did not change differentially in our results, the difference in the content of 2-Deoxy-D-ribose and the expression of *rbsK* under different carbon sources may be another reason for the change in GADP. The strain adjusted the substrate content and enzyme activity, etc., to adjust the specific reaction rate, thereby adapting to different carbon sources. D-glucono-1,5-lactone (GDL) is hydrolyzed by RGN (gluconolactonase) to produce D-Gluconate, which is an important intermediate in sugar metabolism. Then, D-Gluconate is phosphorylated by gluconokinase (gntk) to G6P (D-gluconate-6P), and G6P can be further converted to GAP (D-glycer-aldehyde-3P), which is one of the products of galactose metabolism. In our results, the content of GDL in *ΔVdCreA* and *ΔVdCreC* was lower than that in wild-type strains. Before and after glucose repression, the gene encoding *RGN* was downregulated under starch conditions and upregulated under cellulose conditions. GAP only accumulated in *ΔVdCreA* and *ΔVdCreC* when starch was the carbon source. When cellulose was the carbon source, the content of GAP in the mutant was lower. Although content changes in D-Gluconate and G6P were not detected, the gene encoding gntk that catalyzes its phosphorylation was essentially downregulated after repression. The content of D-sedo-heptulose-7P (S7P) increased in all types of strains after repression. Among the genes encoding enzymes involved in the change in S7P content, the expression of *tktA* (transketolase) and one *talB* (transaldolase) changed synchronously. In addition, in the pentose phosphate pathway, there were also some *PRPS*, *rbsK*, and *talB* genes that were only induced by one type of carbon source. These results reflected the changes in the main pathway activity and key metabolite content of *V. dahliae* during the CCR process. The pentose phosphate pathway underwent significant modification. GDL may be noteworthy as a progenitor to vital intermediates of glucose metabolism, which is markedly transformed in both the genetic and metabolic strata in reaction to both carbon sources.

### 2.12. VdCreA and VdCreC Influence the Expression of Genes Encoding PCWDE

Plant cell wall-degrading enzymes (PCWDE) are commonly associated with fungal pathogenicity. We collected all PCWDE genes from *V. dahliae* based on published data, totaling 100 genes across 22 gene families. We screened the sets of DEGs from different comparative groups (Figure 13A). Regardless of the carbon source, almost all PCWDE genes showed significant differential expression between at least two groups. Under starch conditions, 72 out of 100 PCWDE genes (72%) were identified as DEGs, while under cellulose conditions, 85 (85%) PCWDE genes exhibited differential expression. Among the two largest PCWDE gene families responsible for degrading the pectin main chain, namely polygalacturonase and pectate lyase, 17 and 19 out of the total 22 members, respectively, showed differential expression under starch and cellulose conditions. Further grouping of PCWDEs based on their respective functions and statistical analysis of expression level distribution and median expression within each group focused on those unaffected by glucose repression. Under starch conditions (Figure 13B), despite varying degrees, *ΔVdCreA* led to a decrease in the expression levels of all PCWDE genes, while *ΔVdCreC* resulted in a decrease in the expression levels of PCWDE genes associated with pectin, cellulose, xylan, and cutin, with an increase in the expression levels of PCWDE genes related to galactomannan. Conversely, under cellulose as the carbon source (Figure 13C), there was a noticeable overall increase in the PCWDE expression trend in *ΔVdCreA*, particularly in terms of pectin and cellulose, while a significant decrease was observed in PCWDE expression in *ΔVdCreC*. To elucidate the specific roles of *VdCreA* and *VdCreC*, we selected PCWDE genes that showed a significant strong correlation with the expression of *VdCre* (Pearson correlation coefficient > 0.8, *p* value < 0.05). Under starch conditions (Figure 13D), 19 PCWDE genes were significantly positively correlated with *VdCreA*, while 6 were significantly negatively correlated. Among all 25 significantly correlated genes, 16 (64%) were involved in pectin degradation on the plant cell wall. In total, 17 PCWDE genes were significantly negatively correlated with *VdCreC*, while 4 were significantly positively correlated. Out of all 21 PCWDE genes, 17 (80.95%) were associated with pectin degradation. Significant correlations were found between *VdCreA* and *VdCreC* and a large number of PCWDE genes involved in pectin degradation, with *VdCreA* primarily showing positive correlations and *VdCreC* showing negative correlations. Conversely, under cellulose as the carbon source (Figure 13E), 24 out of 41 PCWDE genes were significantly negatively correlated with *VdCreA*, while 19 were significantly positively correlated with *VdCreC*. Moreover, among the 31 and 20 PCWDE genes correlated with *VdCreA* and *VdCreC*, respectively, 19 (61.29%) and 10 (50%) were associated with pectin degradation. Despite a decrease in the number of PCWDE genes associated with pectin degradation, it still approached or exceeded half of the total genes correlated with *VdCreA* and *VdCreC*. All these results collectively suggest that *VdCre* could influence the expression changes in fungal cell wall-degrading enzyme genes, thereby leading to alterations in the pathogenicity of the strain. This regulatory effect varies under different carbon sources, but consistently exerts a significant impact on pectin degradation-related activities.

## 3. Discussion

Studies have found that CreA has two conserved zinc finger domains, which allow it to repress gene expression by binding directly to the 5′-SYGGRG-3′ motif in the promoters of the target genes [5]. It is therefore proposed that CreA regulates CCR by a double-lock control; that is, CreA physically competes with upstream regulators for DNA binding at the promoter of downstream genes to repress their expression (i.e., block transcriptional activation) [24,25]. However, we found that VdCreA in *V. dahliae* lacks two N-terminal zinc finger domains and part of the nuclear location signal sequences. Transcriptomics revealed *VdCreA* co-expressed with 23 genes (Figure 8A) regardless of whether starch or cellulose was used as the carbon source, and SYGGRG-like motifs were identified in the proximal promoters of the seven co-expressed genes including genes encoding PYCR, PSP1, nsLTPs, two zinc fingers, and two hypothetical proteins. These results suggested that VdCreA could directly regulate the expression of these target genes, and it is speculated that VdCreA may co-regulate the expression of these target genes with other proteins due to VdCreA lacking zinc finger domains.

A recent study revealed that only a small number of genes are actually subjected to the double-lock control, while the expression of the majority of the downstream pathway genes is indirectly regulated by CreA [7]. Our study showed that there were 271 DEGs in *ΔVdCreA* compared with the wild-type strain under repression conditions, indicating these 271 DEGs may be the main genes involved in CCR and VdCreA may indirectly regulate most of these genes. Moreover, there were more TFs in the 271 DEGs under repressing conditions, mainly four *Zn-clus* (Zn2Cys6 TF) and three *GNAT* (Gcn5-related N-acetyltransferases) TFs. Zn-clus is a class of fungus-specific TFs, widely involved in various metabolic processes [26]. *GNAT* belongs to the lysine acetyltransferases enzymes, and it is thought to be mainly responsible for histone acetylation modification [27]. The two groups shared one *SNF2* TF, which is a well-known part of the ATP-dependent chromatin remodelers [28]. This suggests that VdCreA may indirectly regulate the expression of more genes by affecting the expression of these TF genes.

Compared to the wild-type strain V592, the *VdCreA* knockout mutant tends to grow slowly with reduced conidiation and microsclerial production, which is in line with previous studies [13,29,30]. This could be due to the downregulation of some important cellular processes like protein translation or amino acid metabolism by the knockout of *VdCreA*. We found that PYCR, participating in the arginine and proline metabolism pathway, was highly expressed in *ΔVdCreA*, while the proline content in *ΔVdCreA* was extremely low. Similar results were observed in *Staphylococcus aureus*, where PYCR (ProC) under CCR interferes with the synthesis from proline to arginine [31]. This suggests that VdCreA may target the regulation of PYCR expression, thereby mediating the conversion between proline and arginine and affecting the normal growth of the strain.

In this study, we identified and functionally characterized VdCreC, a protein containing a WD40-repeat domain in *V. dahliae*. Our results showed that VdCreC is a key regulator of CCR and also closely involved in conidiation, microsclerial production, and pathogenicity in this fungus. This is consistent with MoCreC in *M. oryzae* [15]. Transcriptomic analyses revealed that *VdCreC* was only strongly directly correlated with 1 gene (*VDAG_09398*) and indirectly correlated with 28 secondary network genes. However, metabolic analysis showed that 84 metabolites were strongly correlated with *VdCreC* expression. These metabolites were involved in the synthesis of various amino acids such as arginine, proline, and phenylalanine or enriched in secondary metabolite synthesis, glycolysis/gluconeogenesis, and folate biosynthesis. This suggests that VdCreC is indirectly implicated in alterations in a multitude of pathways related to carbon metabolism, and it may affect the normal growth of the strain by participating in the synthesis of various amino acids and some secondary metabolites. This is congruent with the role of *VdCreC* in preserving protein stability.

We demonstrated that distinct carbon source conditions induced entirely different gene expression networks in *V. dahliae*. One gene encoding a 6-phosphogluconolactonase served as a hub gene under starch conditions, and five genes encoding G6PD, 2-ODGH, PrpC, MCL, and glyoxylate reductase served as hub genes under cellulose conditions. There was no overlap of genes in the two co-expression networks when starch and cellulose were used as the carbon source; however, the resultant gene functions in the two networks overlapped and their strongly correlated metabolites exhibited similarities. This suggests that *V. dahliae* may complete CCR through similar pathways with different gene expressions. Enrichment analysis showed that these metabolites were widely involved in lipid metabolism, nucleotide metabolism, amino acid metabolism, carbohydrate metabolism, etc., indicating that CCR could systematically affect the activity and efficiency of various biochemical processes, thereby widely mediating the growth and development and physiological metabolism dynamics of *V. dahliae*. This is consistent with previous findings of intricate connections between the regulation of carbon metabolism and diverse cellular functions [8,30].

We have further analyzed the integrated expression profiles of genes and metabolites in the glycolysis/gluconeogenesis and pentose phosphate pathway under different carbon sources and CCR conditions. The results showed that the two major sugar metabolism-related pathways were completely changed. There were 91 DEGs in the glycolysis/gluconeogenesis pathway and 37 DEGs in the pentose phosphate pathway; however, only the contents of 7 and 6 metabolites in the two pathways have been detected to have differential changes, respectively. Among them, GADP changes significantly and is an important metabolite associated with the two pathways, indicating GADP may be a pivotal factor of CCR under different carbon sources. These results imply that the metabolites with significant changes in content may be directly participating in carbon metabolism, thereby affect fungal growth and development. However, the contents of most metabolites in the two pathways were not altered, while the genes encoding the enzymes that catalyze these metabolites had significant differential expression. This disparity in expression networks is potentially instrumental to fungal adaptation to intricate carbon sources. Through the modulation of core genes or nuanced adjustment of peripheral gene expression, fungi can exhibit sufficient flexibility in modulating the CCR process.

It was reported that the *V. dahliae* genome encodes a large number of PCWDEs, and characterization of the exoproteome revealed that at least 52 proteins participate in the pectin and cellulose degradation pathways [32]. All these PCDWEs are responsible for fungal pathogenicity. In this study, we found that both *VdCreA* and *VdCreC* influenced the expression of genes encoding PCDWEs, and their influence depended on different carbon sources. Under starch conditions, *VdCreA* knockout led to a decrease in the expression levels of almost all PCWDE genes (Figure 13B), 19 out of 25 PCWDE genes were significantly positively correlated with *VdCreA*, and 64% were involved in pectin degradation; while under cellulose conditions, there was a noticeable overall increase in the PCWDE expression trend (Figure 13C), 24 out of 41 PCWDE genes were significantly negatively correlated with *VdCreA*, and 61.29% were associated with pectin degradation. Under starch conditions, *VdCreC* knockout resulted in an overall decrease in the expression levels of PCWDE genes, except for an increase in the expression levels of PCWDE genes related to galactomannan (Figure 13B); 17 out of 21 PCWDE genes were significantly negatively correlated with *VdCreC*, and 80.95% were associated with pectin degradation; while under cellulose conditions, there was a significant decrease in PCWDE expression (Figure 13C), 19 out of 41 PCWDE genes were significantly positively correlated with *VdCreC*, and 50% were associated with pectin degradation. In addition, irrespective of the carbon source, *VdCreA* and *VdCreC* either had opposite regulatory effects on the expression of the same PCWDE genes or regulated the expression of different CWDE genes, respectively (Figure 13D,E). These results collectively suggest that *VdCreA* and *VdCreC* may use different regulatory mechanisms to regulate PCWDE genes’ expression, thereby leading to alterations in the pathogenicity of the strain.

## 4. Materials and Methods

### 4.1. Fungal Strains and Growth Conditions

The virulent defoliating strain V592 of *V. dahliae* isolated from an infected cotton plant was used as the wild-type strain [33], and mutants were generated from V592 in this study. To assess their phenotypic characteristics, fungal strains were cultured on potato dextrose agar (PDA: 200 g potato, 15 g glucose, 15 g agar powder) plates at 25 °C in the dark for 15 days or cultured on Czapek–Dox media (CDM: 2 g NaNO_3_, 1 g K_2_HPO_4_, 0.5 g KCl, 0.24 g MgSO_4_, 0.01 g FeSO_4_·7H_2_O) containing different sole carbon sources with or without D-glucose at 25 °C in the dark for 7 days (Constant-Temperature Incubator, MMM Company, Stadlern, Germany). To collect conidia for spore count and infection assays, fungal strains were cultured in liquid Czapek–Dox medium for 7 days at 25 °C with shaking at 150 rpm/min (Intelligent shaker, Tianjin Honour Instrument Co., Ltd., Tianjin, China).

### 4.2. Identification and Phylogenetic Analysis of Target Genes

The mycelia of the V592 strain were collected from PDA medium cultured at 25 °C in the dark for 7 days to extract genomic DNA and total RNA using a fungal DNA Extraction Kit (BIOER, Hangzhou, China) and fungal RNA Kit (Omega Inc., Norwalk, CT, USA) according to the manufacturer’s procedures, respectively. cDNA synthesis was performed with the PrimeScript™ RT reagent kit (TaKaRa, Dalian, China). The genomic DNA and cDNA were used as templates to amplify the full lengths of *VdCreA* and *VdCreC*. All of the primers used in this study are listed in Appendix A. The conserved motifs and protein structure of VdCreA and VdCreC were analyzed by the MEME program (https://meme-suite.org/meme/tools/meme (accessed on 23 October 2023)). The sequences of homologous proteins of CreA and CreC from other fungi were downloaded from the NCBI database (http://www.ncbi.nlm.nih.gov/ (accessed on 23 October 2023)). The phylogenetic analysis was accomplished using MEGA 7.0 via the neighbor-joining method and bootstrap tests replicated 1000 times. The phylogenetic tree was visualized by the Interactive Tree of Life online tool (https://itol.embl.de/ (accessed on 23 October 2023)).

### 4.3. Vector Construction and Fungal Transformation

The *VdCreA* and *VdCreC* single-gene knockout mutants and their complementary strains were obtained by *Agrobacterium tumefaciens*-mediated transformation (ATMT) [34]. To generate the knockout plasmids, upstream and downstream fragments of each target gene were amplified from the genomic DNA of the V592 strain using corresponding primers. Then, the resultant PCR products were fused with linearized pGKO-HPT by using the ClonExpress MultiS One Step Cloning Kit (Vazyme, Nanjing, China) to construct a knockout plasmid of each target gene. To generate the complementary and overexpression plasmids, each intact target gene including the encoding region, native promoter, and terminator was amplified from amplified from the V592 strain, then fused with an *Xba*I/*Bam*HI-linearized p1300-Neo-oLiC-Cas9-TtrpC vector, and then reintroduced to the corresponding gene-knockout strain to generate complementation strains or reintroduced to the V592 strain to generate overexpression strains by ATMT. The primers used for the plasmid constructions are listed in Appendix A.

### 4.4. Phenotype Assays

To determine the effects of each single-gene (*VdCreA* and *VdCreC*) knockout on fungal growth and the utilization of different carbon sources, the fungal mycelium of each strain cultured on PDA medium at 25 °C in the dark for 7 days was inoculated using a sterilized toothpick into the center of Czapek–Dox agar containing different carbon sources (0.5% glucose, 0.5% xylose, 0.5% sucrose, 0.5% xylan, 0.5% raffinose, 0.5% carboxymethyl cellulose, 0.5% starch, 15 g agar) with or without 2% glucose. The growth rate was measured by checking the diameter of each colony. Czapek–Dox agar with 100 g CMC plus 20 mg/mL 2-DOG, and 100 g starch plus 20 mg/mL 2-DOG were used to test the CCR. For assessment of conidial production, 100 µL aliquots of a 10^7^ conidia/mL suspension of each strain were inoculated into Czapek–Dox liquid medium and incubated for 7 days at 25 °C with 150 rpm/min shaking, then the number of conidia were counted under a microscope with a hemocytometer. To observe the hyphal morphology, each fungal strain was inoculated in the center of a PDA plate, sterilized cover glass slides were inserted into the media, and the sample maintained at 25 °C in the dark for 3 days; then, the hyphal morphology on the cover glass slides was observed under a microscope. For assessments of microsclerotial accumulation, 100 µL aliquots of a 10^7^ conidia/mL suspension of each strain were dropped onto minimal medium (MM: 2 g NaNO_3_, 1 g KH_2_PO_4_, 0.5 g KCl, 0.5 g MgSO_4_·7H_2_O, 2 g glucose, 10 mg citric acid, 10 mg ZnSO_4_·7H_2_O, 10 mg FeSO_4_·7H_2_O, 2.6 mg NH_4_Fe(SO_4_)_3_·12H_2_O, 0.5 mg CuSO_4_·5H_2_O, 0.1 mg MnSO_4_·H_2_O, 0.1 mg H_3_BO_3_, 0.1 mg Na_2_MoO_4_·2H_2_O, 15 g agar) plates overlaid with cellophane and incubated in the dark for 15 days at 25 °C (Constant-Temperature Incubator, MMM company, Stadlern, Germany); then, the resulting microsclerotia was scraped off from the cellophane to weigh the wet weight.

### 4.5. Plate Assay of Amylase and Cellulase Activity

The fungal mycelium of each strain cultured on PDA medium at 25 °C in the dark for 7 days was inoculated in the center of Czapek–Dox agar plates containing 0.5% starch, 0.5% carboxymethyl cellulose (CMC), and 0.5% starch plus 2% glucose, and 0.5% CMC plus 2% glucose, respectively. After incubation at 25 °C in the dark for 10 days, amylase activity was visualized using an iodine test, and cellulase activity was visualized by staining with 0.1% Congo red followed by de-staining with 0.7 M NaCl.

### 4.6. Gene Expression Analysis by RT-qPCR

The fungal strains were cultured on PDA medium with different carbon source at 25 °C in the dark for 7 days. The extraction of total RNA of each fungal strain and cDNA synthesis was as described above. qPCR amplification was performed using the PowerUp SYBR Green Master Mix kit (Thermo Fisher, Vilniaus, Lithuania) with the 7500 real-time PCR system. The obtained results were normalized against the expression of the β-tubulin (DQ266153) of *V. dahliae*. The relative expression level was calculated using the 2^−∆∆Ct^ method. The primers used for gene expression are listed in Appendix A.

### 4.7. Pathogenicity Assays

Pathogenicity assays were performed at the three or fourth true leaf stage of upland cotton cultivar Junmian No. 1 using the unimpaired root-dip inoculation method as previously described [33]. The cotton seedlings were inoculated by immersing their root into 200 mL of 10^7^ conidia/mL suspension of each strain for 30 min, then planted in 1/10 MS liquid medium (Coolaber, Beijing, China) and placed in an environmentally controlled chamber with a photoperiod of 16 h of light and 8 h of darkness at 28 °C. Each fungal strain was tested on three pots of cotton seedlings, with 12 seedlings per pot. The disease severity was evaluated by the percent of leaves showing wilt symptoms [35]. The formula of the disease index (DI) value = [Σ(the seedlings of every grade × relative grade)/(total seedlings × 4)] × 100. The infection assays were performed at least three times.

### 4.8. Transcriptomic Determination and Analysis

RNA-seq was performed on the wild-type V592, *ΔVdCreA*, and *ΔVdCreC* strains before and after glucose repression, using starch and cellulose as carbon sources, respectively. The correspondence between labels and the samples were as follows: VD: wild-type strain using starch as a carbon source, VX: wild-type strain using cellulose as a carbon source, VDD: wild-type strain using starch as a carbon source under glucose inhibition, VXD: wild-type strain using cellulose as a carbon source under glucose inhibition, AD: *ΔVdCreA* strain using starch as a carbon source, AX: *ΔVdCreA* strain using cellulose as a carbon source, ADD: *ΔVdCreA* strain using starch as a carbon source under glucose inhibition, AXD: *ΔVdCreA* strain using cellulose as a carbon source under glucose inhibition, CD: *ΔVdCreC* strain using starch as a carbon source, CX: *ΔVdCreC* strain using cellulose as a carbon source, CDD: *ΔVdCreC* strain using starch as a carbon source under glucose inhibition, CXD: *ΔVdCreC* strain using cellulose as a carbon source under glucose inhibition. Three biological replicates were set for each sample group (added suffix labels 1, 2, 3 for distinction). The extraction of total RNA from each sample was identical to the aforementioned method, total RNA was detected for purity, concentration, and integrity using NanoDrop 2000 (Thermo Fisher Scientific, Wilmington, DE, USA) and Agient2100 (Agilent Technologies, Santa Clara, CA, USA). RNA that passed the test underwent PE150 library construction and second-generation sequencing on the Illumina NovaSeq platform, with library construction and sequencing work commissioned to Beijing Biomarker Biotechnology Co., Ltd., Beijing, China.

The raw reads after sequencing were first filtered for low quality and adapter sequences, and then the filtered sequences were aligned to the *V. dahliae* reference genome (ASM15067v2) using HISAT2. The expression of unigenes was calculated and normalized by FPKM, and differential screening was performed through EBseq2. Those with a fold-change greater than 2 and a false discovery rate (FDR) less than 0.01 were considered differentially expressed genes (DEGs). De novo annotation of unigenes was performed through eggNOG-Mapper (v5), and Kyoto Encyclopedia of Genes and Genomes (KEGG) enrichment analysis was performed through the R package clusterProfiler (*p* value < 0.05). PCA analysis were performed using the R package (PCAtools). In addition, Tbtools (V2) and R (V4) were used for the visualization of all heat maps, box plots, violin plots, and up-set diagrams. All DEGs were first filtered based on expression and missing degree, and then WGCNA was performed through the R package (WGCNA), calculating the scale-free topological fit index and mean connectivity within 1–30 and selecting the optimal soft threshold. Hierarchical clustering of DEGs was performed through the topological overlap measure (TOM), and the similarity module merge threshold was set to 0.25. The Spearman correlation coefficient between genes and metabolites was calculated through R, and the correlation network was visualized by Cytoscape (V3). The identification of consistent sequences in the promoter region was accomplished through MEME-chip, and the sequences of SYGGRG and CGGG motif were compared with the identified motif.

### 4.9. Untargeted Metabolite Detection

The samples used for RNA-seq were also used for untargeted metabolite detection, which was completed by Beijing Biomarker Biotechnology Co., Ltd. Specifically, the samples were dissolved in 0.1% formic acid aqueous solution (mobile phase A) and 0.1% formic acid acetonitrile (mobile phase B). Based on the LC/MS system of a Waters Acquity I-Class PLUS ultra-high performance liquid in tandem with a Waters Xevo G2-XS QT high-resolution mass spectrometer (Waters, Corporation, Milford, MA, USA), dual-channel data collection was performed for low collision energy and high collision energy. The mobile phase A used a gradient of 98%–2%–2%–98%, running for 0.25 min, 10 min, 13 min, and 13.1 min, while maintaining a flow rate of 400 μL/min throughout. The low collision energy was 2 V, and the high collision energy range was 10–40 V. The scanning frequency of the mass spectrometer was 0.2 s. The parameters of the ESI ion source were as follows: capillary voltage: 2000 V (positive ion mode) or −1500 V (negative ion mode); cone voltage: 30 V; ion source temperature: 150 °C; desolvent gas temperature: 500 °C; backflush gas flow rate: 50 L/h; desolventizing gas flow rate: 800 L/h. The chromatographic column was a Waters Acquity UPLC HSS T3 (1.8 μm, 2.1 × 100 mm) (Waters, Corporation, MA, USA). The original data were collected through MassLynx (v4.2) and then peak extraction and alignment by Progenesis QI (V1). Metabolite identification was completed through the METLIN database, with mass deviation controlled within 100 ppm. The functions of all metabolites were annotated using the KEGG, HDBM (human metabolome database), and Lipidmaps (lipid metabolites and pathways strategy) databases. OPLS-DA modeling was applied through the R package ropls, and a permutation test (n = 200) was used to verify the stability of the model. The VIP value of metabolites was calculated through multiple cross-validation, and the differential multiples (greater than 1), *p* value (less than 0.05), and VIP (greater than 1) of the OPLS-DA model were used to screen differentially expressed metabolites (DEMs). The Spearman correlation and KEGG enrichment were consistent with RNA-seq.

### 4.10. Data Statistical Analysis

SPSS 26.0 software was used for statistical analysis of the data. Duncan’s new multiple range method was used to test the significance of the difference. ** represents a significant difference at *p* < 0.01 from Student’s *t*-test.

## Figures and Tables

**Figure 1 ijms-25-11575-f001:**
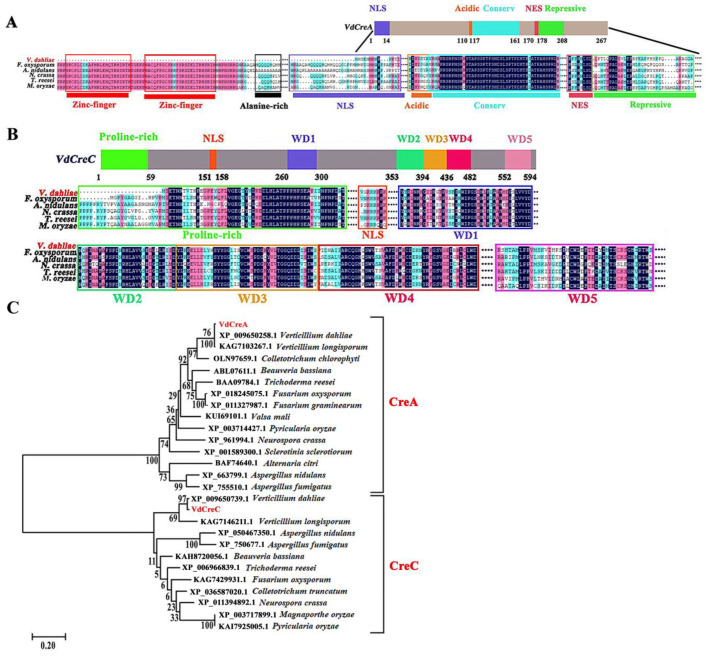
Structure prediction, amino acid sequence alignment, and phylogenetic tree of VdCreA and VdCreC in *V. dahliae*. (**A**) Structure schematic diagrams of the VdCreA protein and amino acid sequence alignment between the VdCreA protein and CreA from other fungi. (**B**) Structure schematic diagrams of the VdCreC protein and amino acid sequence alignment between the VdCreC protein and CreC from other fungi. (**C**) Phylogenetic tree of VdCreA, VdCreC, and homologs from several other fungi constructed by MEGA11 using the neighbor-joining method; the bootstrap test was repeated 1000 times.

**Figure 2 ijms-25-11575-f002:**
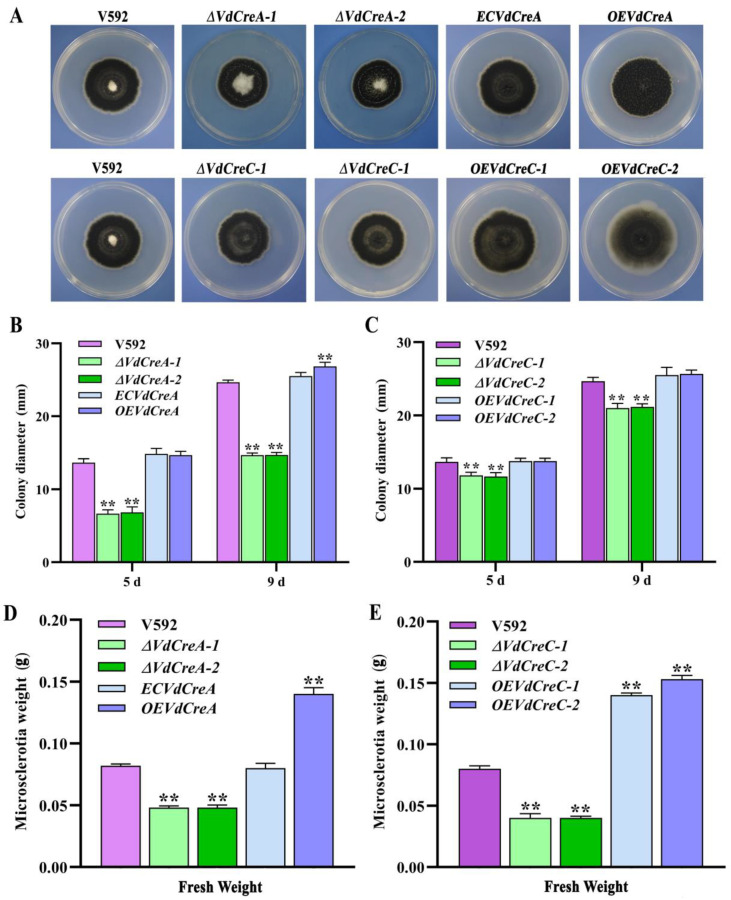
Phenotype characteristics of *VdCreA* and *VdCreC* knockout mutants. (**A**) Colony morphologies of the V592 strain and all *ΔVdCreA* and *ΔVdCreC* mutants grown on PDA medium at 15 dpi. (**B**) Colony diameter of the V592 strain, *ΔVdCreA*, the complementary strain *ECVdCreA*, and the overexpression strain *OEVdCreA* grown on PDA medium at 5 dpi and 9 dpi. (**C**) Colony diameter of the V592 strain, *ΔVdCreC*, and the overexpression strains *OEVdCreC* grown on PDA medium at 5 dpi and 9 dpi. (**D**) The wet weight of the microsclerotia produced by the V592 strain, *ΔVdCreA*, *ECVdCreA*, and *OEVdCreA*. (**E**) The wet weight of the microsclerotia produced by the V592 strain, *ΔVdCreC*, and *OEVdCreC*. Error bars represent standard errors calculated from three replicates. ** represents a significant difference at the 0.01 probability level between the knockout mutants, the complementary strain, the overexpression strains, and the wild-type strain V592 (*p* < 0.01).

**Figure 3 ijms-25-11575-f003:**
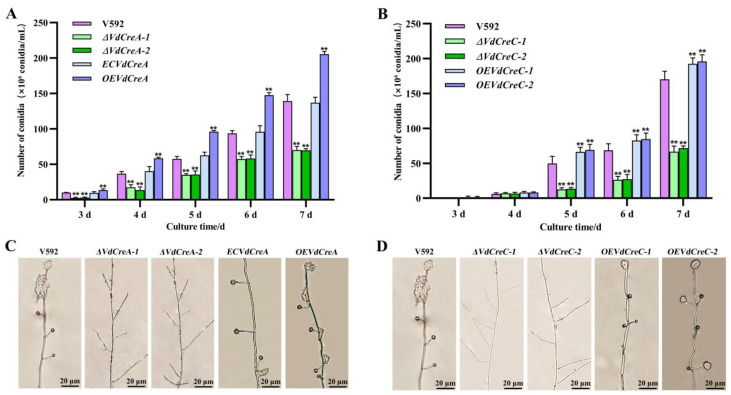
The effect of *VdCreA* and *VdCreC* knockout on conidial production. (**A**) Conidia produced by the V592 strain, *ΔVdCreA*, *ECVdCreA*, and *OEVdCreA*. (**B**) Conidia produced by the V592 strain, *ΔVdCreC*, and *OEVdCreC*. (**C**) Microscopic observation of conidiophore stalks produced by the V592 strain, *ΔVdCreA*, *ECVdCreA*, and *OEVdCreA*. (**D**) Microscopic observation of conidiophore stalks produced by the V592 strain, *ΔVdCreC*, and *OEVdCreC*. Error bars represent the standard deviation. ** represents a significant difference at the 0.01 probability level between the knockout mutants, the complementary strain, the overexpression strains, and the wild-type strain V592 (*p* < 0.01).

**Figure 4 ijms-25-11575-f004:**
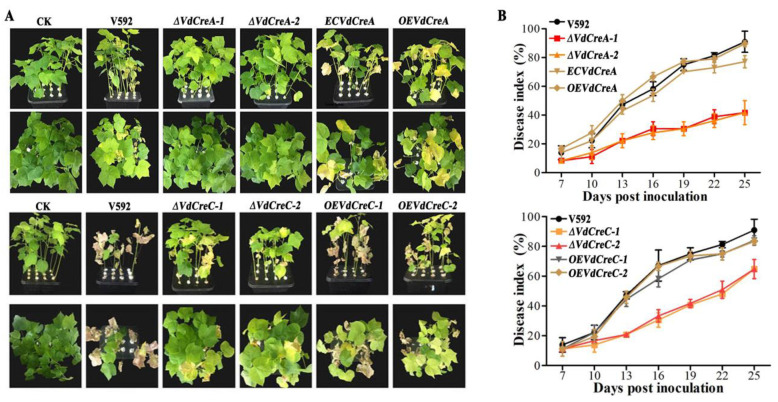
The determination of virulence of different knockout mutant strains to cotton seedlings. (**A**) Disease symptoms of cotton plants infected with different strains at 25 days post inoculation (dpi). (**B**) Disease index of cotton plants infected with different strains. The error bars represent the standard errors calculated from 3 replicates.

**Figure 5 ijms-25-11575-f005:**
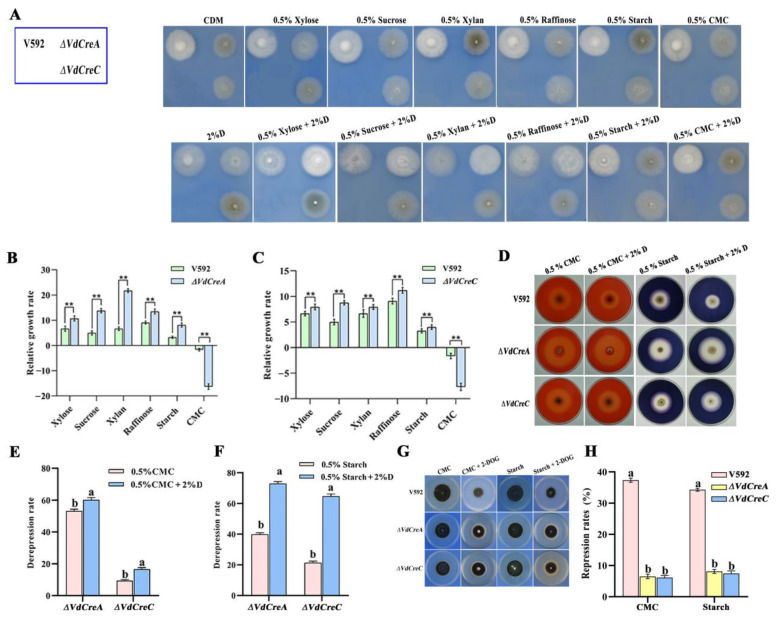
*VdCreA* and *VdCreC* are essential for carbon catabolite repression. (**A**) Phenotypic characteristics of the all strains grown on Czapek–Dox agar plates containing different sole carbon sources with or without glucose. (**B**) The relative growth rate of *ΔVdCreA*. (**C**) The relative growth rate of *ΔVdCreC*. The relative growth rate = (the diameter on carbon sources plus glucose − the diameter on carbon sources)/(the diameter on carbon sources plus glucose) × 100%. (**D**) The amylase and cellulase production of *ΔVdCreA* and *ΔVdCreC* grown on Czapek–Dox agar plates containing starch or CMC with or without glucose. (**E**) The de-repression rate of cellulase activity in *ΔVdCreA* and *ΔVdCreC* grown on CMC and on CMC plus glucose. De-repression rate = (the sizes of clear zone of the knockout mutant − the size of clear zones of V592 strain)/(the size of clear zones of the knockout mutant) × 100%. (**F**) The de-repression rate of amylase activity in *ΔVdCreA* and *ΔVdCreC* grown on starch and on starch plus glucose. (**G**) Colony growth rates of the *ΔVdCreA* and *ΔVdCreC* grown on MM medium containing CMC and starch with or without 2-DOG. (**H**) The repression rates of the *ΔVdCreA* and *ΔVdCreC* mutants. Repression rate = (the diameter of untreated strain − the diameter of treated strain)/(the diameter of untreated strain) × 100%. ** represents a significant difference at the 0.01 probability level between the knockout mutants, the overexpression strains, and the wild-type strain V592 (*p* < 0.01). Different letters above the error bars indicate statistically significant differences at *p* < 0.01 from one-way ANOVA tests.

**Figure 6 ijms-25-11575-f006:**
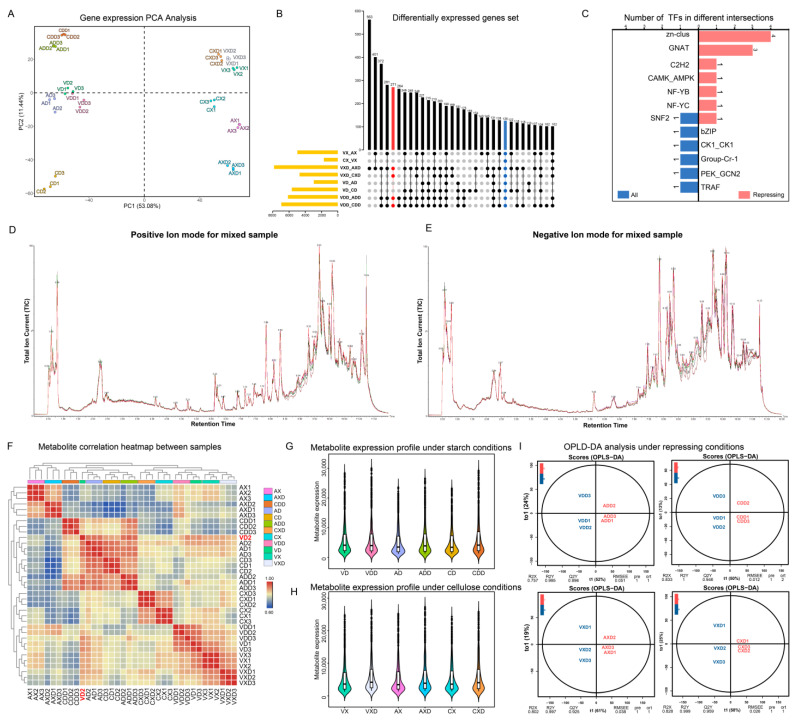
Global statistics of transcriptomic and metabolomic data. (**A**) PCA of the expression levels of all samples. (**B**) Up-set diagram of DEGs in differential groups. Red annotations represent differential groups under repressing conditions, while blue annotations represent all differential groups. (**C**) Statistics of transcription factors in common DEGs under different groups. (**D**,**E**) The total ion current of the mixed sample. (**F**) Heatmap of sample correlation based on metabolite expression levels. Outlier samples are marked in red. (**G**) Boxplot statistics of dominant metabolites under starch conditions. (**H**) Boxplot statistics of dominant metabolites under cellulose conditions. (**I**) OPLS-DA mode diagram of different groups under repressing conditions.

**Figure 7 ijms-25-11575-f007:**
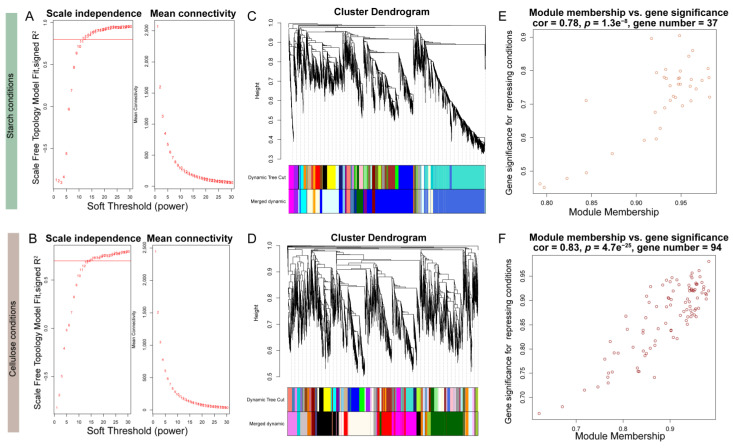
WGCNA analysis in starch and cellulose conditions. (**A**,**B**) Soft-thresholding selection diagram. The red horizontal line represents a threshold of 75%. (**C**,**D**) Gene tree, module division, and merging results. (**E**,**F**) Scatter plot of module correlation with repressing conditions.

**Figure 8 ijms-25-11575-f008:**
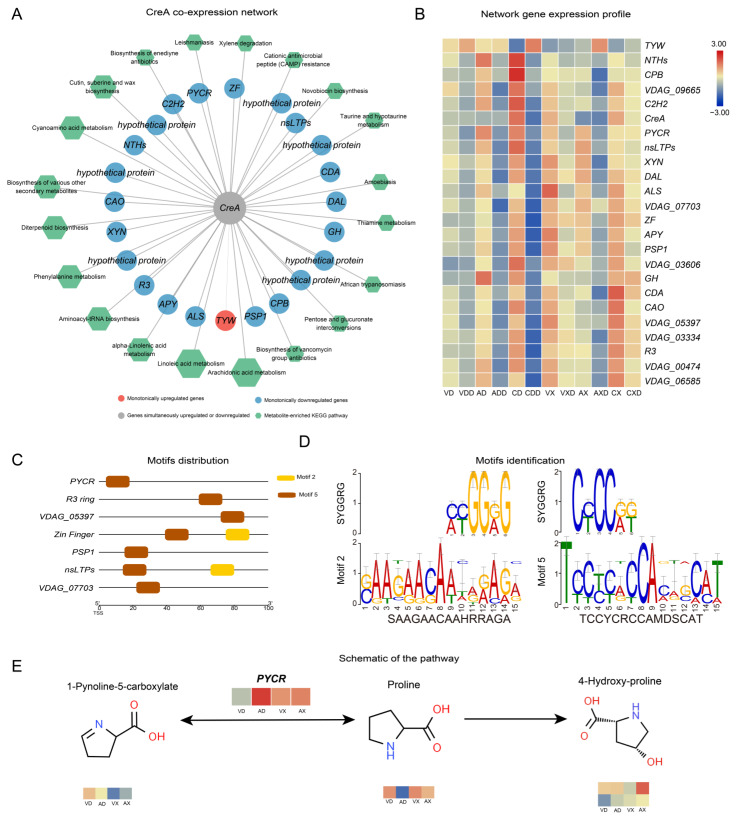
*VdCreA* may target the regulation of PYCR expression, affecting the normal growth of the strain. (**A**) Co-expression network related to the *VdCreA* gene. The circular nodes represent genes, and the color of the circles represents the expression trend of the gene. Hexagonal nodes represent the pathways enriched with metabolites in the network. (**B**) Expression heatmap of genes in the co-expression network: the redder the color, the higher the expression level, and the bluer the color, the lower the expression level. (**C**) Motif 2 and motif 5 are distributed in the proximal promoters of seven co-expressed genes. (**D**) The identification of motif 2 and motif 5. (**E**) The changes in metabolites in the PYCR pathway.

**Figure 9 ijms-25-11575-f009:**
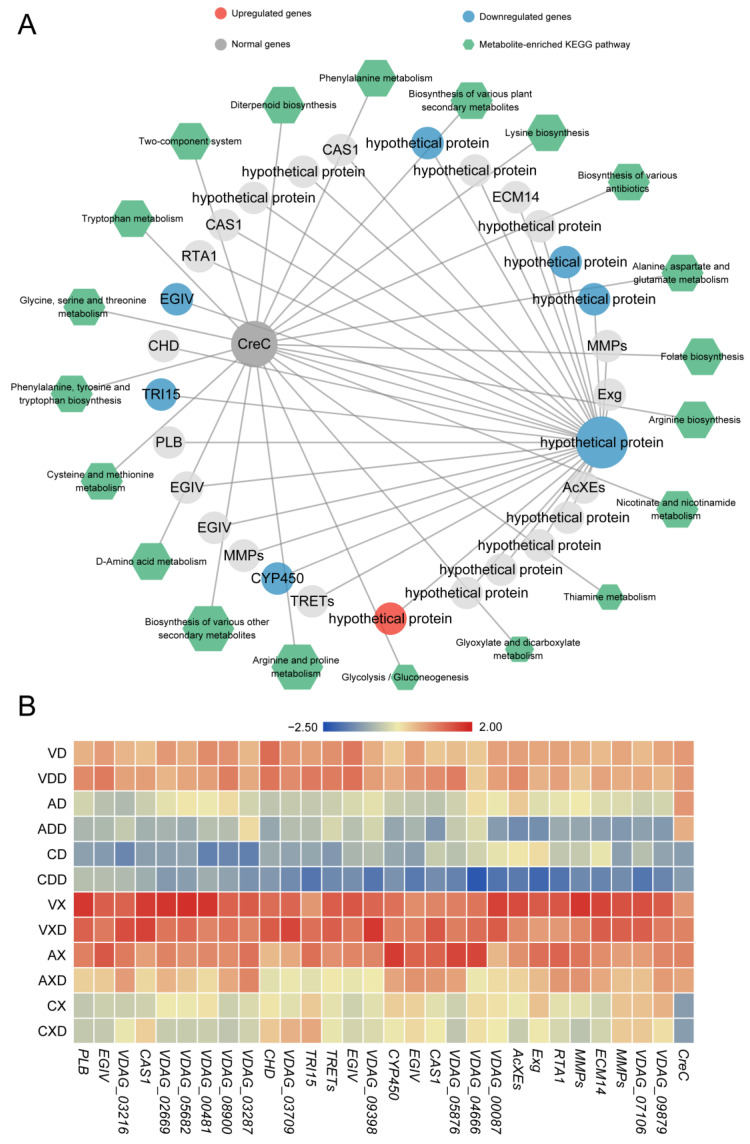
Co-expression network related to repressing conditions. (**A**) Co-expression network related to the *VdCreC* gene. The circular nodes represent genes, and the color of the circles represents the expression trend of the gene. Hexagonal nodes represent the pathways enriched with metabolites in the network. (**B**) Expression heatmap of genes in the co-expression network: the redder the color, the higher the expression level, and the bluer the color, the lower the expression level.

**Figure 10 ijms-25-11575-f010:**
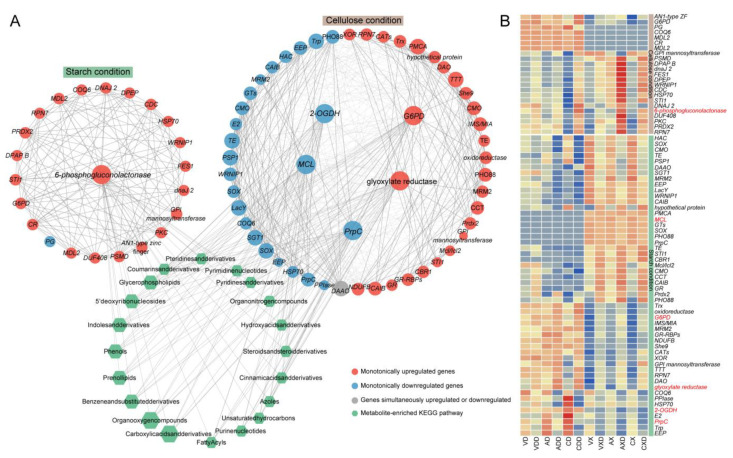
Distinct carbon sources induce entirely different expression networks. (**A**) The co-expression network of genes and metabolites within the module, with genes involved in carbon metabolism serving as hub genes. The circular nodes represent genes, and the color of the circles represents the expression trend of the gene. Hexagonal nodes represent the pathways enriched with metabolites in the network. (**B**) Expression heatmap of genes in the co-expression network and metabolites: the redder the color, the higher the expression level, and the bluer the color, the lower the expression level. The labels of hub genes are highlighted in red.

**Figure 11 ijms-25-11575-f011:**
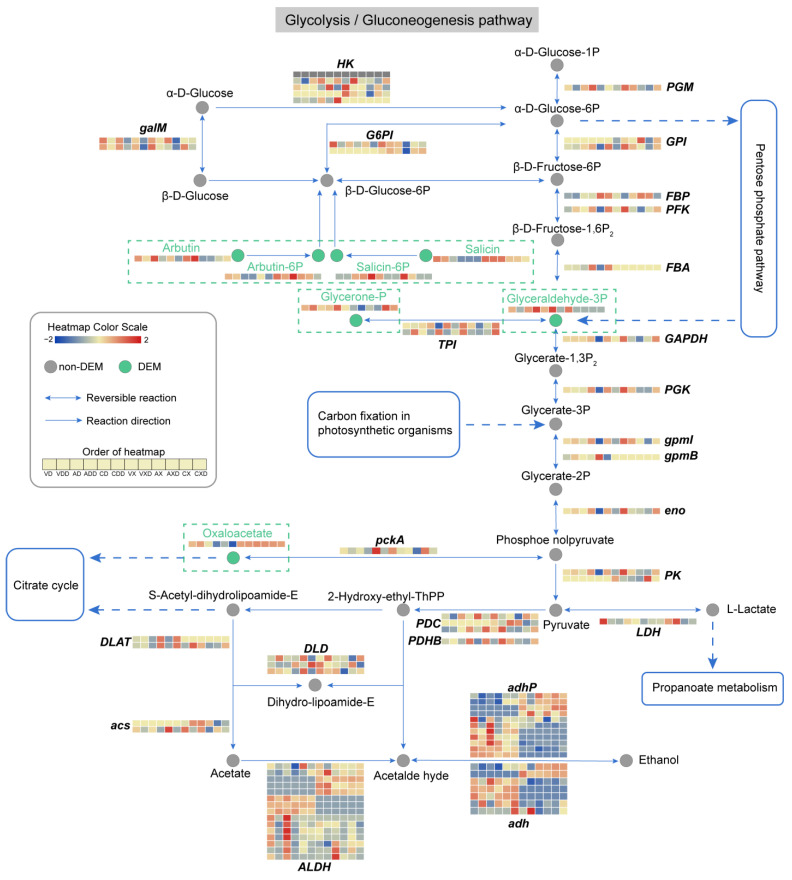
Analysis of the glycolysis/gluconeogenesis metabolic pathway. Circles represent metabolites, gray circles represent non-DEMs, green circles represent DEMs, which are highlighted in the dashed rectangular box for emphasis. Arrows indicate the direction of reaction, and the genes encoding the enzymes catalyzing the reaction are displayed at the corresponding positions. The dashed arrows indicate connections to other related metabolic pathways. The heatmap represents the expression level of genes or metabolites.

**Figure 12 ijms-25-11575-f012:**
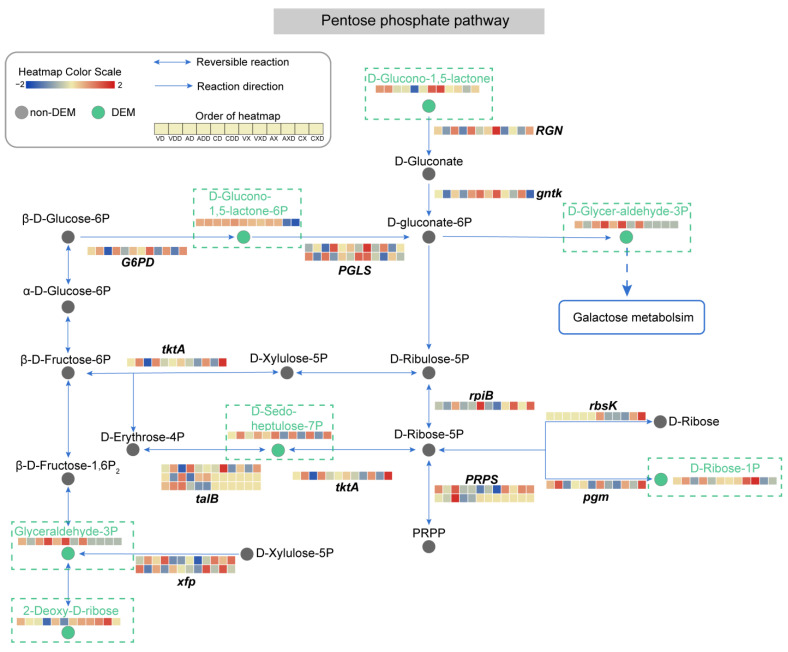
Analysis of the pentose phosphate metabolic pathway. Circles represent metabolites, gray circles represent non-DEMs, green circles represent DEMs, which are highlighted in the dashed rectangular box for emphasis. Arrows indicate the direction of reaction, and the genes encoding the enzymes catalyzing the reaction are displayed at the corresponding positions. The dashed arrows indicate connections to other related metabolic pathways. The heatmap represents the expression level of genes or metabolites.

**Figure 13 ijms-25-11575-f013:**
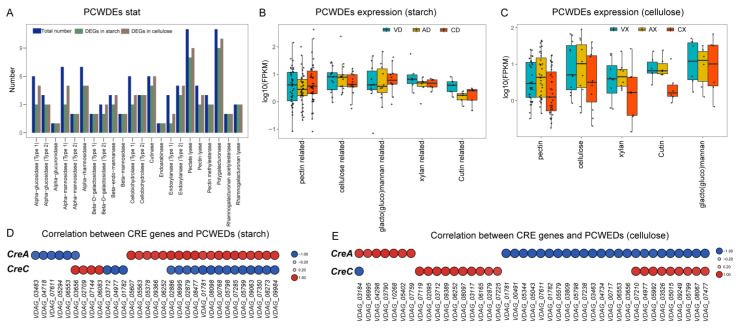
Both *VdCreA* and *VdCreC* influence the expression of fungal cell wall-degrading enzyme genes depending on different carbon sources. (**A**) Bar graph illustrating the statistics of 22 PCWDE gene families. The x-axis represents the gene family names, and the corresponding bars indicate the total number of PCWDE genes in each family and the number of differentially expressed genes under starch conditions and under cellulose conditions. (**B**) Box plots depicting the expression levels of PCWDE genes under starch conditions. All expression levels were normalized by taking logarithms. Different colors represent samples from different groups, and the median expression level is depicted by a black solid line within each box plot. (**C**) Box plots illustrating the expression levels of PCWDE genes under cellulose conditions, with normalization by logarithm. Similar to (**B**), different colors denote samples from different groups, and the median expression level is indicated by a black solid line within each box plot. (**D**) Heat map depicting the correlation between *VdCre* genes and PCWDE genes under starch conditions. All correlation coefficients are greater than 80% with a *p* value less than 0.05. Negative correlations are represented in blue, while positive correlations are shown in red. Larger circles in the heat map indicate higher correlation values. (**E**) Heat map representing the correlation between *VdCre* genes and PCWDE genes under cellulose conditions. Similar to (**D**), all correlation coefficients exceed 80% with a *p* value less than 0.05. Negative correlations are depicted in blue, and positive correlations are displayed in red. Larger circles in the heat map signify higher correlation values.

## Data Availability

Raw sequencing data can be accessed through the Gene Expression Omnibus with the accession number (the data are being audited).

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
