# Peer review of "Combined Transcriptome and Metabolome Analysis Reveals That Carbon Catabolite Repression Governs Growth and Pathogenicity in *Verticillium dahliae"

_ijms, 2024, doi:10.3390/ijms252111575_

Round 1

Reviewer 1 Report

Comments and Suggestions for Authors

1- Pg27, line 824: What are the chromatographic conditions, flow rate, run time, flush time? Equilibrium time? gradient style etc?

 2- Pg27, line 830: what are the collision energies and other details?

3- Fig 13, b & c: in this figure author mentioned box plots, while in figure 6: e & f, author mentioned violin plots. So its better to mention these by using same style.

4- As author mentioned that they identified 2,986 metabolites, so what are the main class of metabolites, are these primary or secondary class? And which are the most significant metabolites among all?

5- Its much better to mention the LCMS profile with characteristic signals.

6- Author need to explain in detail that, which data base and software used for all of these studies.

Author Response

Comment 1: [1- Pg27, line 824: What are the chromatographic conditions, flow rate, run time, flush time? Equilibrium time? gradient style etc?]

Response 1: Thank you for point this out. We have added a description of the method details in lines 871-873.

Comment 2: [2- g27, line 830: what are the collision energies and other details?]

Response 2: Thank you for point this out. We have updated the description of the metabolomics detection methods in lines 873-877, adding key details.

Comment 3: [Fig 13, b & c: in this figure author mentioned box plots, while in figure 6: e & f, author mentioned violin plots. So its better to mention these by using same style.]

Response 3: Thank you for point this out. the violin plots in Fig. 6 represent the expression distribution of major metabolites across all samples, with the number of metabolites reaching over a thousand. They exhibit a rich frequency distribution. In contrast, Fig. 13 shows the expression distribution of PCWD genes, where different classes of related genes number only in the range of several to a few dozen. When visualizing this with violin plots, due to the small data size, there is not enough frequency distribution to support significant graphical variations. As shown below, the violin plots may appear very close to box plots, resulting in a less favorable visual effect. Therefore, we believe that box plots are more appropriate in this context. If you insist on your viewpoint, we can change the graphical representation in further revisions.

Schematic representation of the violin plot in Fig. 13 of the manuscript

Comment 4: [As author mentioned that they identified 2,986 metabolites, so what are the main class of metabolites, are these primary or secondary class? And which are the most significant metabolites among all?]

Response 4: Thank you for point this out. We have added a description of metabolite annotation and classification in lines 291-297.

Comment 5: [Its much better to mention the LCMS profile with characteristic signals.]

Response 5: Dear Reviewer, we sincerely apologize for not clearly understanding your comment. In all sections of the manuscript related to metabolites, we have used the "expression" of metabolites as the basis for analysis, which is a common method in all such articles. We are unclear about what "LCMS profile with characteristic signals" specifically refers to. Should we change the method used for analysis? Or supplement certain values for specific metabolites? We apologize again and hope you can provide more direct and detailed comments. Once we understand this issue, we will promptly update it in further revisions.

Comment 6: [Author need to explain in detail that, which data base and software used for all of these studies.]

Response 6: We reviewed the Materials and Methods section and added descriptions of the software and databases used in lines 851-853, 881-883 and 890-893. In our view, the main methods employed in the study should now be comprehensive.

Reviewer 2 Report

Comments and Suggestions for Authors

The manuscript (ID ijms-3180441) submitted for review expands the current knowledge and introduces new knowledge regarding the studied problem. In my opinion, it fits very well into the scope of current molecular studies of phytopathological problems. On the one hand, the summary is extensive, but on the other hand it contains important information regarding the research carried out - so it can be left in its current form. I also have no comments on the introduction, which is well written and contains the latest knowledge on the subject. The authors also did a good job of presenting their research results. However, I have a few comments about this part of the manuscript: 1) the text contains minor errors, e.g. ln 109 "neurospora crassa" - the name should be capitalized. I ask the authors to re-read the text carefully and eliminate all errors. 2) Figs. are generally very well developed conceptually, but their resolution/size makes them difficult to read - or maybe their quality has decreased during file conversion? Therefore, please improve the readability of Figs. The discussion is conducted in a thoughtful way and is complete. However, similarly to the "Results" chapter, I have noticed several minor shortcomings, e.g. Ln 602 “domains[9, 10],”. So I the authors to re-read the text carefully and eliminate all errors. Materials and methods are written in a clear and understandable way. However, they lack some information that would help to reproduce the experiments 100%. Therefore, please supplement this section with the following information: 1) add the accuracy of the devices, e.g. used for inubation 2) add the composition or manufacturer's name of the culture media used, e.g. a note regarding PDA. In summary, in my opinion, the reviewed manuscript is written in a logical and coherent manner and contains interesting research results. Therefore, after correction and improvement, it will be an interesting work that will be worthy of publication in International Journal of Molecular Sciences.

Author Response

Comment 1: [the text contains minor errors, e.g. ln 109 "neurospora crassa" - the name should be capitalized. I ask the authors to re-read the text carefully and eliminate all errors.]

Response 1: Thank you for point this out. We have revised the clerical error, and we have re-read the the text carefully and eliminate several errors and Highlight these revision.

Comment 2: [Figs. are generally very well developed conceptually, but their resolution/size makes them difficult to read - or maybe their quality has decreased during file conversion? Therefore, please improve the readability of Figs.]

Response 2: We have re-uploaded all images in PDF and TIFF (300 dpi) formats, and we have comprehensively updated the visual quality of the images, increasing the font size. All images should now be clear and easy to read.

Comment 3: [However, similarly to the "Results" chapter, I have noticed several minor shortcomings, e.g. Ln 602 “domains[9, 10],”. So I the authors to re-read the text carefully and eliminate all errors.]

Response 3: We have revised the clerical error, and we have re-read the the text carefully and eliminate several errors and Highlight these revision.

Comment 4: [Materials and methods are written in a clear and understandable way. However, they lack some information that would help to reproduce the experiments 100%. Therefore, please supplement this section with the following information: 1) add the accuracy of the devices, e.g. used for inubation 2) add the composition or manufacturer's name of the culture media used, e.g. a note regarding PDA.]

Response 4: Thank you for point this out. We have added the descriptions of the devices used for fungal culture in lines 738, for fungal reproduction in lines 741. We have also added the descriptions of the composition of the culture media in lines 735-739, in lines 788-791, and in line 815.

Round 2

Reviewer 1 Report

Comments and Suggestions for Authors

Manuscript is nicely written. Kindly add LCMS chromatograms in the main manuscript and Mass m/z values of few important secondary metabolites as supplementary file.

Author Response

Comment 1: Manuscript is nicely written. Kindly add LCMS chromatograms in the main manuscript and Mass m/z values of few important secondary metabolites as supplementary file.

Response 1: We have added the TIC plot of the mixed sample in Figure6 and uploaded the mass spectral information and annotation results of all metabolites in the supplementary files Table S2. Accordingly, we have included descriptions of the relevant results in the revised manuscript in lines 291-302 and lines 327-328. We hope these updates meet your expectations, and we thank you again for your help in improving our manuscript.

Round 3

Reviewer 1 Report

Comments and Suggestions for Authors

Accept